# Collagen Family and Other Matrix Remodeling Proteins Identified by Bioinformatics Analysis as Hub Genes Involved in Gastric Cancer Progression and Prognosis

**DOI:** 10.3390/ijms23063214

**Published:** 2022-03-16

**Authors:** Mihaela Chivu-Economescu, Laura G. Necula, Lilia Matei, Denisa Dragu, Coralia Bleotu, Andrei Sorop, Vlad Herlea, Simona Dima, Irinel Popescu, Carmen C. Diaconu

**Affiliations:** 1Department of Cellular and Molecular Pathology, Stefan S. Nicolau Institute of Virology, 030304 Bucharest, Romania; laura_dragomir2006@yahoo.com (L.G.N.); liliamatei@yahoo.com (L.M.); denisa_dragu81@yahoo.com (D.D.); cbleotu@yahoo.com (C.B.); ccdiaconu@yahoo.com (C.C.D.); 2Faculty of Medicine, Titu Maiorescu University, 040441 Bucharest, Romania; herlea2002@yahoo.com (V.H.); irinel.popescu220@gmail.com (I.P.); 3Center of Excellence for Translational Medicine, Fundeni Clinical Institute, 022328 Bucharest, Romania; sorop_andrei@yahoo.com (A.S.); dima.simona@gmail.com (S.D.); 4Department of Pathology, Fundeni Clinical Institute, 022328 Bucharest, Romania; 5Center of General Surgery and Liver Transplantation, Fundeni Clinical Institute, 022328 Bucharest, Romania

**Keywords:** collagens, extracellular matrix, cell adhesion, biomarkers, survival, targeted therapy

## Abstract

Gastric cancer has remained in the top five cancers for over ten years, both in terms of incidence and mortality due to the shortage of biomarkers for disease follow-up and effective therapies. Aiming to fill this gap, we performed a bioinformatics assessment on our data and two additional GEO microarray profiles, followed by a deep analysis of the 40 differentially expressed genes identified. PPI network analysis and MCODE plug-in pointed out nine upregulated hub genes coding for proteins from the collagen family (COL12A1, COL5A2, and COL10A1) or involved in the assembly (BGN) or degradation of collagens (CTHRC1), and also associated with cell adhesion (THBS2 and SPP1) and extracellular matrix degradation (FAP, SULF1). Those genes were highly upregulated at the mRNA and protein level, the increase being correlated with pathological T stages. The high expression of BGN (*p* = 8 × 10^−12^), THBS2 (*p* = 1.2 × 10^−6^), CTHRC1 (*p* = 1.1 × 10^−4^), SULF1 (*p* = 3.8 × 10^−4^), COL5A1 (*p* = 1.3 × 10^−4^), COL10A1 (*p* = 5.7 × 10^−4^), COL12A1 (*p* = 2 × 10^−3^) correlated with poor overall survival and an immune infiltrate based especially on immunosuppressive M2 macrophages (*p*-value range 4.82 × 10^−7^–1.63 × 10^−13^). Our results emphasize that these genes could be candidate biomarkers for GC progression and prognosis and new therapeutic targets.

## 1. Introduction

Gastric cancer remains one of the top five cancers both in terms of incidence and mortality. According to GLOBOCAN 2020’s latest data, gastric cancer (GC) is the fifth most common cancer worldwide, with over 1,000,000 new cases each year, and also the fourth cause of cancer-related death in the world, with a 5-year survival rate of around 32% [1]. The high rate of mortality is due to the fact that the majority of GC patients are diagnosed at late stages when treatment is often useless (about 20% of GC patients are diagnosed in early stages in Europe), and the therapeutic approach still consists only of combined chemo-radiation therapy followed by surgical resection [2]. A significant percentage of GC patients develop recurrent disease represented by distant metastases, despite surgery and perioperative treatment [3]. However, there are no effective biomarkers for GC diagnosis. The lack of biomarkers for GC diagnosis represents an important issue in the management of this malignancy. Tumor markers such as alpha-fetoprotein (AFP), carcinoembryonic antigen (CEA), CA19–9, CA72.4, and CA125 have low sensitivity 34 (<40%), and their specificities are modest for GC patients [4]. HER2 overexpression was recognized as another biomarker in GC, and its presence was documented to be associated with 12–22% of GC [5,6]. HER2 positive GC was correlated with more aggressive disease and poor outcomes [7]. These tumors respond to therapies that include trastuzumab, but the improvement of median overall survival was modest, by only 2.7 months, according to the results from the ToGA trial [8]. Therefore, significant efforts are being made to improve the clinical management of GC through the discovery of new biomarkers and shaping up of new technologies for cancer management.

Substantial advances in cancer biology have resulted from bioinformatics analysis, which assist in the identification of significantly deregulated genes and highlight the pathogenic pathways involved in cancer development and progression. In recent years, a wide range of bioinformatics methods that have combined multiple databases of biological information such as the Gene Expression Omnibus (GEO), protein–protein interaction (PPI) network, STRING tool, and Cytoscape software have led to certain breakthroughs. As follows, several key genes signatures were reported to be involved in GC development and prognosis. Thus, Wang et al. reported FN1, COL1A1, INHBA, and CST1 to be associated with worse overall survival in GC [9], Lu et al. found that SPP1 and FN1 were correlated with tumor relapse and poor prognosis [10], Liu et al. identified TIMP1, SPP1, CXCL8, THY1, and COL1A1 genes to be negatively correlated with survival [11], and Chong et al. found FN1, TIMP1, SPP1, APOE, and VCAN genes to be associated with poor overall survival in GC patients [12]. Although TIMP1 is a tissue inhibitor of metalloproteinases, it may also have MMP-independent functions in solid cancers. TIMP-1 can stimulate cell proliferation, accelerating tumor invasion and metastasis, via important signaling pathways such as NOTCH and WNT [13,14], with recent reports demonstrating the poor prognostic value of TIMP-1-positive expression in solid cancers [15]. Interestingly, some of the genes highlighted by these meta-analyses appeared repeatedly, such as FN1, SPP1, TIMP1, or collagen, most of which are involved in tissue reorganization.

The novelty of our results consists of emphasizing the role of the collagen family and of other proteins associated with the assembly mechanism of collagen fibers and with their degradation. The results indicate the important role that extracellular matrix (ECM) reorganization plays during the carcinogenesis process. To this end, we used our previous data GSE103236, together with other two gene expression microarray datasets from the Gene Expression Omnibus (GEO) (GSE13911, GSE79973), and several bioinformatics tools, to identify aberrant expressed genes significantly involved in gastric carcinogenesis and progression. Moreover, we used correlation with immune infiltrate and survival analyses to explore the prognostic value of the selected genes.

We consider that this study could provide a potential progression and prognosis biomarkers panel for GC and new therapeutic targets for cancer management.

## 2. Materials and Methods

### 2.1. Microarray Dataset Information

To identify commonly deregulated genes (DEGs) in GC, our previous data GSE103236 [16], and two additional gene expression microarray data were downloaded from GEO (https://www.ncbi.nlm.nih.gov/geo/; access date 5 November 2021) [17], GSE13911 [18], and GSE79973 [19], including a total of 55 gastric tumor samples and 53 adjacent non-tumor tissues, were used for differential analysis. Gene expression was analyzed by the GEO2R tool, with log FC ≥ 2 and *p* < 0.05 as standards to identify DEGs. The Benjamini and Hochberg false discovery rate method was applied for Geo2R analyses. After that, the common DEGs profile was obtained through a Venn diagram (http://bioinformatics.psb.urgent.be/webtools/Venn/; access date 6 October 2021).

### 2.2. Common DEGs Enrichment Analysis

Common DEGs lists were analyzed using the Database for Annotation, Visualization and Integrated Discovery (DAVID) 6.8 (https://david.ncifcrf.gov/; access date 8 October 2021) [20] and Kyoto Encyclopedia of Gene and Genome (KEGG). Gene ontology (GO) enrichment was used to identify molecular function (MF), cellular component (CC), and biological process (BP), and KEGG to pinpoint the main affected pathways.

### 2.3. Protein–Protein Interaction (Ppi) Network Analysis

PPI network was achieved using STRING (https://string-db.org/; access date 12 October 2021) tool [21]. The analysis assessed the correlation between protein products. Then, Cytotype Molecular Complex Detection (MCODE) plug-in in Cytoscape software was used to detect hub genes in the PPI network [22]. The MCODE parameters were set as follows: node score cut-off = 0.2, k-core = 2, maximum depth from seed = 100.

### 2.4. Gene Expression Analysis

The expression levels of hub genes in different pathological stages of GC were assessed in UALCAN web tool (http://ualcan.path.uab.edu/; access date 9 November 2021), based on TCGA online available RNAseq data. Differential mRNA expression analysis includes 351 gastric tumor samples (18 in stage I, 123 in stage II, 169 in stage 3, and 41 in stage IV) and 34 normal tissues. The *p*-value for Student’s *t*-test was set as follows * *p* < 0.05, ** *p* < 0.01, *** *p* < 0.001.

### 2.5. Patient and Specimens

Pairs of human gastric tumor and adjacent normal tissues were collected from GC patients during surgery at the Center of General Surgery and Liver Transplantation of Fundeni Clinical Institute, after written informed consents and approval of the Fundeni Clinical Institute Ethical Committee (No 52495/2018). Tissue samples from tumor and adjacent tissue from the proximal resection margin were selected by pathologists and frozen in liquid nitrogen immediately after excision and stored at −80 °C. The GC samples were classified according to the American Joint Committee on Cancer TNM (tumor, node, and metastasis) staging. None of the patients had received preoperative chemotherapy or radiotherapy.

### 2.6. Western Blot Analysis

Whole protein extracts were obtained using RIPA buffer supplemented with Complete O, Mini, EDTA-free Protease Inhibitor Cocktail (Roche Applied Science, Penzberg, Germany) and quantified using BCA Protein Assay kit (Pierce, Rockford, IL, USA). Proteins (40 µg) were electrophoretically separated by SDS-PAGE and transferred onto PVDF membranes. The membranes were blocked in Tris-Buffer Saline (TBS)-0.5% Tween 20 with 2% bovine serum albumin, and then incubated with the primary antibodies against proteins of interest at 4 °C overnight. Used antibodies (1:1000 dilutions) were: mouse monoclonal anti-human biglycan (BGN) antibody (MAB2667, R&D, Minneapolis, MN, USA), sheep polyclonal anti-human fibroblast activation protein alpha (FAP) antibody (-AF3715, R&D, Minneapolis, MN, USA), rabbit-monoclonal anti-human collagen type X alpha 1 (COL10A1) antibody (NBP2-66988, Novus Biologicals, Littleton, CO, USA), and anti-beta actin monoclonal antibody (AC-15) (AM4302, Invitrogen, Carlsbad, CA, USA). Secondary antibodies used were: anti-mouse (HAF007, R&D, Minneapolis, MN, USA), anti-rabbit (HAF008, R&D, Minneapolis, MN, USA), and anti-sheep (HAF016, R&D, Minneapolis, MN, USA), all conjugated with HRP. Signals were developed using ECL HRP chemiluminescent substrate (WP20005, Invitrogen) and captured using MicroChemi 4.2 system (Bio Imaging Systems).

### 2.7. Kaplan–Meier Plotter Database Analysis

The correlations between the expression of selected genes and overall survival (OS) of GC were tested in the Kaplan–Meier plotter (http://kmplot.com/analysis/; access date 22 October 2021) [23].

### 2.8. TIMER

TIMER2.0 (https://cistrome.shinyapps.io/timer/; access date 17 January 2022) [24] was used to analyze the relationship between the hub genes and 12 immune cell types in the tumor microenvironment. A heat map with Spearman’s rho was generated presenting the correlation of the expression of selected genes with various immune cells in GC.

## 3. Results

### 3.1. Identification of DEGs in GC Tissue

Three independent microarray studies deposited in GEO OMNIBUS were selected for comparison of all genes to identify genes linked to GC pathogenesis. A schematic representation for methods applied during analysis is presented in the Appendix A (Appendix A). We used GEO2R online tool to select upregulated and downregulated genes for each subset, and then the obtained profiles were run through the Venn diagram tool. Results revealed 22 common upregulated genes and 18 common downregulated genes (Figure 1). Gene’s names are detailed in Table 1.

### 3.2. Functional Enrichment Analysis of Common DEGs

We used DAVID to perform a biological functions enrichment analysis of common DEGs. When analyzing the upregulated genes, we identified clusters of genes involved in cell adhesion, collagen catabolic process, extracellular matrix organization, and collagen fibril organization (Figure 2).

The main pathways involved were ECM (extracellular matrix)–receptor interaction, protein digestion, focal adhesion, and PI3K-Akt signaling. The main terms that appeared during analysis were secretion, ECM, cell adhesion, and collagen.

### 3.3. PPI Network Enrichment and Cytoscape Analysis of the Module Genes and Hub Genes

Next, we analyzed the interaction between upregulated DEGs using PPI network analysis and applied the MCODE plug-in in Cytoscape to obtain hub genes. The PPI enrichment *p*-value for the upregulated DEGs network was <1.0 × 10^16^, showing a high interaction between genes that are most probably biologically connected, as a group (Figure 3A). The highest interconnectivity was observed for three genes from the collagen family: collagen type 1, 10, and 12 alpha 1 chain (COL1A1, COL10A1, and COL12A1) and Thrombospondin-2 (THBS2).

Network string interaction was then analyzed in Cytoscape for hub genes based on the MCODE score. The results identified nine upregulated hub genes including CTHRC1, BGN, FAP, THBS2, COL12A1, COL5A2, SULF1, SPP1, and COL10A1 (Figure 3B). Most of the hub genes are coding for collagen family proteins (COL12A1, COL5A2, COL10A1), negative regulators of collagen matrix deposition (collagen triple helix repeat-containing protein 1—CTHRC1), and other proteins involved in collagen fiber assembly (biglycan—BGN). Other genes are coding for proteins involved in extracellular matrix degradation (Prolyl endopeptidase—FAP and sulfatase 1—SULF1), cell to cell and cell to matrix interactions (Thrombospondin-2—THBS2, Osteopontin—SPP1) (Table 2).

### 3.4. The Expression Levels of Hub Genes in GC

We used UALCAN to analyze the mRNA expression level of hub genes in GC samples from different stages and normal adjacent tissue. The analysis was performed on TCGA samples that include 351 gastric tumor samples (18 in stage I, 123 in stage II, 169 in stage 3, and 41 in stage IV) and 34 normal tissues. The results showed that the expression levels of all hub genes in stages 2, 3, and 4 were significantly higher than normal tissues (*p* < 0.05) (Figure 4A). Moreover, CTHRC1, BGN, and FAP also displayed a higher expression in stage 1 than normal tissues. Importantly, the expression levels of all nine genes in stage 2, 3, and 4 were significantly higher than those in stage 1, except for SPP1 (Figure 4A). Overall, the increase in expression levels of all hub genes was correlated with pathological stages in GC.

Next, protein expression was tested in clinical specimens of tumor and normal gastric tissue through western blot assay (Figure 4B). Results showed an increase in COL10A1 protein in tumor tissue, and in BGN and FAP proteins involved in collagen fibril assembly and matrix degradation, respectively, emphasizing that these hub genes could have a crucial function in gastric tumorigenesis.

The results show an abundance of ECM proteins, such as collagens, and remodeling enzymes (BGN, FAP) in tumor tissue, compared to the normal tissue. These proteins are secreted mainly by cancer-associated fibroblast and infiltrating immune cells.

### 3.5. Relation between Selected DEGs and Clinical Outcome

To assess the prognostic value of selected DEGs in GC, specific survival curves were generated using the Kaplan–Meier plotter (Figure 5). Results showed that high expression of BGN (*p* = 1.2 × 10^−8^), COL5A1 (*p* = 1.3 × 10^−4^), COL10A1 (*p* = 5.7 × 10^−4^), COL12A1 (*p* = 2 × 10^−3^), CTHRC1 (*p* = 1.1 × 10^−4^), SULF1 (*p* = 3.8 × 10^−4^), and THBS2 (1.2 × 10^−6^) are significantly correlated with poor overall survival and may be effective prognostic biomarkers for GC. Among them, BGN, THBS2, and CTHRC1 have a higher hazard ratio (HR) for poor overall survival then other hub genes, 1.68 and 1.55, respectively.

### 3.6. TIMER Analysis

To find out if the hub genes are associated with the inflammatory response, and therefore are influencing the poor survival of GC, we used the TIMER tool. The analysis showed that our hub genes, associated with alterations in the extracellular matrix and cell adhesion, are negatively correlated with the abundance in B cells, CD8+ T cells, CD4+ Th1 cells, T regulatory (Tregs) cells, and activated dendritic cells (Figure 6). However, a positive and significant correlation was noticed with the presence of macrophages, the predominant type being the immunosuppressive M2 macrophages (R value 0.255–0.448, *p*-value range 4.82 × 10^7^–1.63 × 10^13^), which predicts a poor prognosis (Appendix A).

## 4. Discussion

The key findings of our study are nine hub genes related to the collagen family, assembly and cell adhesion, which are upregulated during gastric carcinogenesis and tumor progression according to the mRNA and protein level analysis, and highly biologically connected according to the PPI network and MCODE plug-in analyses. Furthermore, the high expression of these genes was related to poor overall survival according to Kaplan–Meier plotter. This result is sustained by TIMER analysis that showed that upregulation of these genes was positive and significantly associated with the presence of an immune infiltrate based mostly on tumor-associated macrophages, especially on immunosuppressive M2 macrophages.

One of the reasons why GC remains as one of the top five cancers, both in terms of incidence and mortality, is the lack of effective biomarkers. In recent years, several bioinformatics studies have emerged emphasizing the importance of the collagen family in solid cancer development. Thus, Chen Y et al. identified COL1A1, COL1A2, and COL12A1 as prognostic biomarkers and immune-associated targets in GC using two GEO OMBIBUS data files with 25 pairs of gastric tumor and adjacent non-tumor mucosa tissues [25]. Moreover, Zhaoxing Li et al. extended the list including other family members such as COL1A1, COL1A2, COL3A1, COL5A2, COL4A1, FN1, COL5A1, COL4A2, and COL6A3, where COL1A1 and COL1A2 were proposed as poor prognostic biomarkers for GC [26].

Our study, involving our own data and two additional GEO profiles including a total of 55 gastric tumor samples and 53 adjacent non-tumor tissues, identified a common list containing 40 DEGs. Further analysis including PPI network analysis and MCODE plug-in in Cytoscape pointed out nine upregulated hub genes including CTHRC1, BGN, FAP, THBS2, COL10A1, COL12A1, COL5A2, SULF1, and SPP1. These upregulated genes are involved in several processes that are associated with carcinogenesis, such as modulation of cell adhesion (THBS2 and SPP1), collagen fibril organization (COL12A1, COL5A2, and COL10A1, BGN), collagen catabolic process (CTHRC1), and ECM degradation (FAP, SULF1). They belong to signaling pathways that sustain ECM–receptor interaction, protein digestion, focal adhesion, and PI3K-Akt signaling.

Collagen family proteins, together with elastins, fibronectins, and laminins, play an important role in tissue organization as parts of the ECM, sustaining tissue resistance and its main form [27]. During wound repair, the microenvironment tries to limit the tumor by attracting an inflammatory infiltrate, which, through the secreted cytokines, recruit fibroblasts that close the wound, and, in the end, will secrete MMPs that will remodel the collagen matrix, allowing wound resolution. In cancer, there is a disturbance in the balance between synthesis and protein degradation in the ECM, which has the effect of remodeling the matrix [28]. Tumor fibrosis (desmoplasia) is characterized by chronic inflammation and high numbers of cancer-associated fibroblasts that secrete abundant ECM proteins, such as collagens, and remodeling enzymes that reorganize and strengthen the matrix. Moreover, cancer-associated cells, via the secreted factors (IL-6), influence the immune response towards a pro-tumor phenotype, attracting pro-tumorigenic immune cell infiltrate (M2 macrophage, Th2 cells, Tregs, etc.). In the meantime, tumor cells begin to secrete matrix-degrading enzymes, which in turn degrade the matrix and release cytokines and growth factors that signal cancer cells proliferation, favoring tumor growth and progression [29].

Over time, it has become increasingly clear that desmoplasia is compromising cancer treatment, playing an active role in therapeutic resistance, and, therefore, in cancer progression. There are also evidences that collagens together with fibronectin, integrin, and laminin, and other components of ECM, are directly involved in tumor initiation and progression to metastasis by engaging in the EMT program by inducing signals through focal adhesion kinase, a core component of integrin signaling, promoting ERK and PI3K signaling pathways [30,31]. More and more data are accumulating supporting the collagen effects on surrounding tumor cells, where they are directly regulating cell proliferation, differentiation, gene expression, migration, invasion, metastasis, and survival [32,33,34,35].

A similar degradation of the extracellular matrix is found in some genetic diseases (e.g., Ehlers–Danlos syndrome) associated with mutations in collagen genes. Thus, a recent study looked at the incidence of mutations in collagen genes and their role in gastric tumor progression, as well as their association with survival. The results showed that the mutations were associated with a distinctive lower matrisome expression, due to the loss of collagen expression and secretion, strongly associated with improved outcomes [36].

The identified genes were further confirmed to be highly upregulated in gastric cancer samples compared to normal tissue, the mRNA expression being correlated with the increase in tumor T stage on the TCGA samples by UALCAN analysis. Our results on protein expression using western blot assay also showed an increase in COL10A1, BGN, and FAP proteins in tumor tissue compared with adjacent normal tissue, consistent with staging.

The novelty of our results consists of emphasizing the role of the collagen family and of other proteins associated with the assembly mechanism of collagen fibers and with their degradation. The results indicate the important roles that degradation of the structure and normal functioning of the ECM play in the carcinogenesis process.

The collagen family, including COL10A1, COL12A1, and COL5A2, was reported to be overexpressed in various types of epithelial cancers, including GC. Recent studies associate these molecules with processes such as migration, invasion, and poor overall survival. Moreover, inhibition of the gene expression reduces cell proliferation and invasion [34,37,38,39,40,41,42]. The overexpression of biglycan (BGN) was also identified in GC and was associated with poor prognosis, while inhibition of BGN enhanced chemotherapeutic efficacy. BGN was found to be secreted by tumor endothelial cells and was able to induce tumor angiogenesis and metastasis [43]. Fibroblast activation protein α (FAP), a protein involved in tissue remodeling, can sustain invasion of the adjacent tissue in cancer, and was reported to be overexpressed in colorectal cancer being correlated with survival [44]. Cancer-associated fibroblasts that express FAP show immunosuppressive proprieties for the tumor microenvironment [45]. Collagen triple helix repeat-containing (CTHRC1) was considered a cancer-related factor that sustains migration processes, proliferation, invasion, and metastasis in GC. Moreover, it is considered that CTHRC1 could promote early-stage cancer and is a candidate as a prognostic biomarker, signaling tumor recurrence or metastasis [46]. An increased expression of THBS2 seems to sustain cancer progression in GC [47] and is associated with a poor prognosis in colorectal cancer [48]. Moreover, a recent study reported circulating THBS2 and CA19-9 levels as possible candidates for a panel that detects early stages of pancreatic ductal adenocarcinoma [49]. Secreted phosphoprotein 1 (SPP1) expression level was correlated with tumor stage and aggressiveness in several cancers, including colorectal cancer [50,51]. SULF1 (human sulfatase 1) is overexpressed in GC [52] and the inhibition of SULF1 expression resulted in decreased proliferation, migration, and invasion in urothelial carcinomas cell lines [53].

Our findings suggest that these genes could be candidate biomarkers for GC progression. Since many of the identified genes can be measured through the soluble plasma circulating proteins with available immune-enzymatic tests, these biomarkers offer a less invasive and more accessible approach to obtain real-time progression information of the tumor burden and have proved to be really helpful in estimating overall survival [54]. These blood-based biomarkers may be particularly beneficial in monitoring the disease progression during tumor therapeutic management since repeated tissue biopsies are difficult to obtain and cause real distress to the patient.

Collagen fibers and matrix remodeling enzyme correlation with tumor aggression and immune infiltrate can be exploited to predict cancer patient outcome. Kaplan–Meier analysis revealed a significant correlation between the high expression of BGN (*p* = 1.2 × 10^8^), COL5A1 (*p* = 1.3 × 10^4^), COL10A1 (*p* = 5.7 × 10^4^), COL12A1 (*p* = 2 × 10^3^), CTHRC1 (*p* = 1.1 × 10^4^), SULF1 (*p* = 3.8 × 10^4^), and THBS2 (1.2 × 10^6^) and poor overall survival.

An explanation for the negative impact of hub gene overexpression on survival may come from the analysis of the type of inflammatory infiltrate. The TIMER analysis showed that upregulation of these genes was positively correlated with M2 macrophages that are associated with more aggressive tumor features, reflected by tumor progression, invasion, and metastasis [55,56]. M2 macrophages create an immunosuppressive microenvironment, which favors angiogenesis by directly secreting vascular epithelial growth factors (VEGFs), and various immunosuppressive molecules such as TGFb, IL-10, or immune checkpoints [57,58]. Quite the opposite to this, M1 macrophages have pro-inflammatory activities and high antigen-presenting capacity, being very important in the fight against tumor cells. The ratio of M1/M2 is currently being used in assessing tumor prognosis. Polarization toward the M2 phenotype, reflected by low M1/M2 ratio was found to be a predictor for poor prognosis in several cancers [59,60].

The tumor-promoting inflammatory infiltrate is recruited and activated by tumor stroma [61,62]. This is due to an early programing during cancer development of cancer-associated fibroblast, via IL-1β-secretion and nuclear factor-κB (NF-κB) activation, to sustain a tumor-promoting inflammatory response [63]. Acerbi I et al. demonstrated that the stroma of the invasive region of the most aggressive Basal-like and Her2 breast tumor subtype, was rich in collagen fibers and stiffened, and also presented the greatest number of infiltrating M2 macrophages and the highest level of TGF beta. These findings indicate that cancer progression, collagen deposition, and matrix stiffness are linked, and implicate tissue inflammation and TGF beta [64]. The M2 infiltrate was associated with poor prognosis in colorectal or oral squamous epithelial cells [65,66].

Cancer-associated fibroblasts can reduce the activation of various immune effector cells, such as cytotoxic CD8+ T cells and natural killer (NK) cells, by expressing immune checkpoints inhibitory molecules such as programmed death ligands PD-L1, PD-L2, and anti-cytotoxic T lymphocyte-associated protein 4 CTLA-4 [67]. In this way, cancer-associated stroma can influence the tumor immunity, favoring a pro-tumorigenic tumor microenvironment. Additionally, studies have shown that M2 macrophages can directly inhibit the T cell response by expressing PD-L1 on their surface [68]. This aspect favors the efficacy of anti-PD-1/PD-L1 or anti-CTLA-4 therapy [69]. Furthermore, some studies on animal models have shown that macrophage blockade by inhibiting colony stimulating factor 1 receptor (CSF-1R), which controls the production, differentiation, and function of macrophages, can add further value to immune checkpoint blockade therapy [69]. As a result of these preliminary research, the idea of combining immune checkpoint blockade therapy with CSF-1R antagonists has been applied in clinical trials (e.g., NCY02323191).

Numerous ongoing studies are dedicated to finding an inhibitor for tumor-associated fibrosis. Starting from the observation that collagen-producing myofibroblasts express on their surface fibroblast activation protein (FAP), a FAP-targeted PI3K/mTOR inhibitor that specifically targets FAP-expressing myofibroblasts was recently developed. The inhibitor effectively reduced collagen production, showing that collagen-producing cells could be an effective target in human lung fibrosis [70].

Clinical trials combining immunotherapy, targeted therapy, and chemotherapy are in progress, and will represent a landmark in cancer management [71].

## 5. Conclusions

The overexpressions of hub genes identified in our study, mainly associated with changes in the extracellular matrix and cell adhesion, have been shown to be important biomarkers that predict poor prognosis in GC, especially due to the association with an inflammatory infiltrate composed mainly of M2 macrophages with an inhibitory effect on the activation of T lymphocytes. Moreover, the study highlighted the significant value of collagen family members for the development of new targeted therapies that may be associated with immune checkpoint blockade therapy and CSF-1R inhibitors.

## Figures and Tables

**Figure 1 ijms-23-03214-f001:**
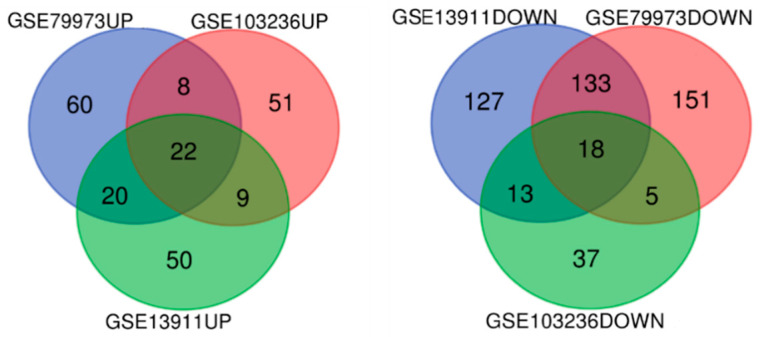
Venn diagrams of upregulated and downregulated genes in all three GC microarray datasets.

**Figure 2 ijms-23-03214-f002:**
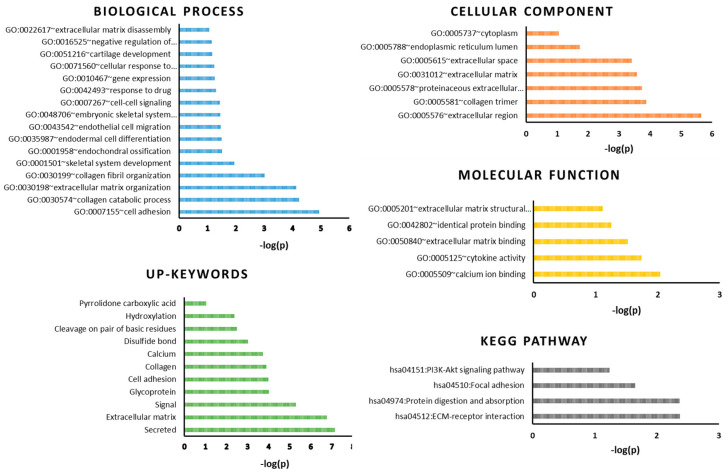
DAVID gene ontology enrichment analysis for common DEGs performed for biological process, cellular component, molecular function, significant keywords, and KEEG pathway.

**Figure 3 ijms-23-03214-f003:**
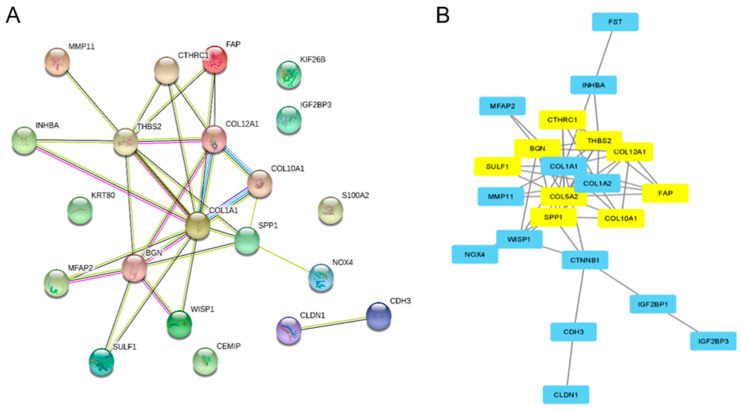
Protein–protein interaction (PPI) analysis. (**A**) PPI networks of the common upregulated DEGs. (**B**) Identification of nine hub genes according to MCODE score.

**Figure 4 ijms-23-03214-f004:**
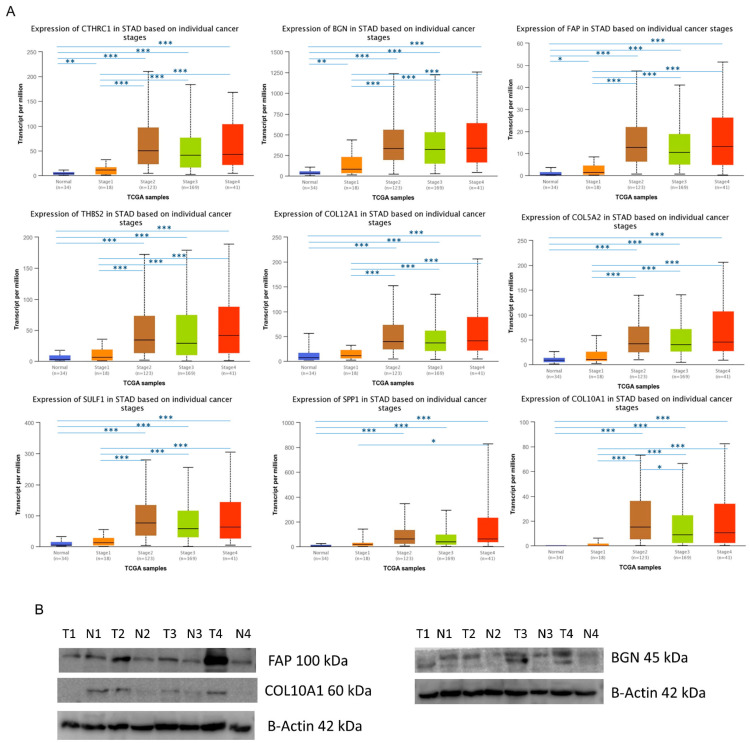
The expression levels of hub genes in GC. (**A**) The mRNA expression was correlated with pathological T stages. The asterisk represents the comparison between a specific stage and the normal group, or between different stages. * *p* < 0.05, ** *p* < 0.01, *** *p* < 0.001. (**B**) Representative western blotting for COL10A1, BGN, and FAP proteins in 4 paired gastric tumor (T) and adjacent non-tumor tissues (N) with progressive stages from T1 to T4.

**Figure 5 ijms-23-03214-f005:**
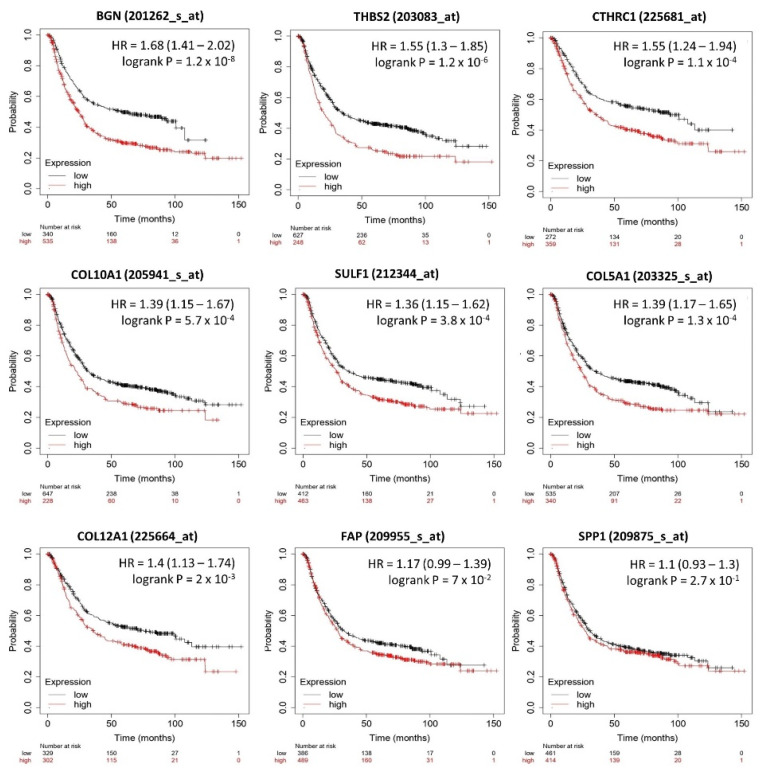
Overall survival curves according to high and low DEG expression. High gene expression has a higher hazard ratio (HR) for poor overall survival.

**Figure 6 ijms-23-03214-f006:**
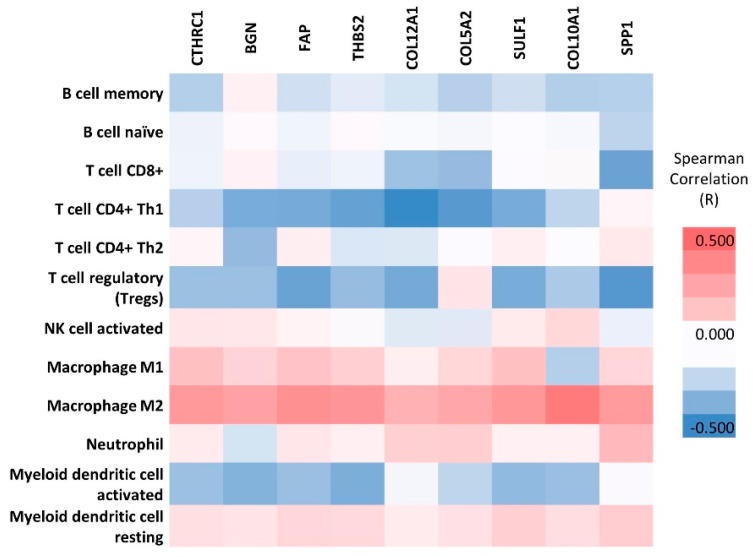
Analysis of tumor infiltrate in GC. Correlation between hub genes and 12 immune cell types in tumor microenvironment based on Spearman’s rho.

**Table 1 ijms-23-03214-t001:** Common upregulated and downregulated genes.

Upregulated Genes (Symbol)	Gene Name
CDH3	cadherin 3
KIF26B	kinesin family member 26B
CEMIP	cell migration inducing hyaluronan binding protein
CTHRC1	collagen triple helix repeat-containing 1
IGF2BP3	insulin like growth factor 2 mRNA binding protein 3
SULF1	sulfatase 1
KRT80	keratin 80
FAP	fibroblast activation protein alpha
THBS2	thrombospondin 2
BGN	biglycan
INHBA	inhibin beta A subunit
S100A2	S100 calcium binding protein A2
SPP1	secreted phosphoprotein 1
MFAP2	microfibrillar associated protein 2
COL1A1	collagen type I alpha 1 chain
WISP1	WNT1 inducible signaling pathway protein 1
COL12A1	collagen type XII alpha 1 chain
CLDN1	claudin 1
NOX4	NADPH oxidase 4
COL10A1	collagen type X alpha 1 chain
MMP11	matrix metallopeptidase 11
IL11	interleukin 11
**Downregulated Genes (Symbol)**	**Gene Name**
GC	GC, vitamin D binding protein
AKR1C1	aldo-keto reductase family 1 member C1
SCARA5	scavenger receptor class A member 5
TPCN2	two pore segment channel 2
SLC2A12	solute carrier family 2 members 12
LIFR	leukemia inhibitory factor receptor alpha
SIGLEC11	sialic acid binding Ig like lectin 11
FGA	fibrinogen alpha chain
ATP4A	ATPase H+/K+ transporting alpha subunit
CKMT2	creatine kinase, mitochondrial 2
CCKBR	cholecystokinin B receptor
GHRL	ghrelin/obestatin prepropeptide
GIF	gastric intrinsic factor
ATP4B	ATPase H+/K+ transporting beta subunit
MAL	mal, T cell differentiation protein
CHGA	chromogranin A
ESRRG	estrogen related receptor gamma
SST	somatostatin

**Table 2 ijms-23-03214-t002:** Hub genes selected by Cytoscape MCODE score and their biological role.

No.	Gene Name	MCODE Score	Annotation and Biological Role
1	CTHRC1	5	Collagen triple helix repeat-containing protein 1; may act as a negative regulator of collagen matrix deposition.
2	BGN	5	Biglycan; may be involved in collagen fiber assembly; small leucine rich repeat proteoglycans.
3	FAP	5	Prolyl endopeptidase FAP; cell surface glycoprotein serine protease that participates in extracellular matrix degradation and is involved in many cellular processes including tissue remodeling, fibrosis, wound healing, inflammation, and tumor growth. Both plasma membrane and soluble forms exhibit post-proline cleaving endopeptidase activity, with a marked preference for Ala/Ser-Gly-Pro-Ser/Asn/Ala consensus sequences, on a substrate such as alpha-2-antiplasmin SERPINF2 and SPRY2. Degrades also gelatin, heat-denatured type I collagen, but not native collagen type I and IV, vibronectin, etc.
4	THBS2	4.46	Thrombospondin-2; adhesive glycoprotein that mediates cell to cell and cell to matrix interactions. Ligand for CD36 mediating antiangiogenic properties
5	COL12A1	4.46	Collagen alpha-1(XII) chain; type XII collagen interacts with type I collagen containing fibrils, the COL1 domain could be associated with the surface of the fibrils, and the COL2 and NC3 domains may be localized in the perifibrillar matrix; belongs to the fibril-associated collagens with interrupted helices (FACIT) family.
6	COL5A2	4.46	Collagen alpha-2(V) chain; type V collagen is a member of group I collagen (fibrillar forming collagen). It is a minor connective tissue component of nearly ubiquitous distribution. Type V collagen binds to DNA, heparan sulfate, thrombospondin, heparin, and insulin. Type V collagen is a key determinant in the assembly of tissue- specific matrices (by similarity).
7	SULF1	4	Extracellular sulfatase Sulf-1. Modifies the structure of heparan sulfate chains, an important component of the extracellular matrix, and, thereby, alters the function of the ECM.
8	SPP1	4	Osteopontin; Binds tightly to hydroxyapatite. Appears to form an integral part of the mineralized matrix. Probably important to cell–matrix interaction; endogenous ligands.
9	COL10A1	4	Collagen alpha-1(X) chain; type X collagen is a product of hypertrophic chondrocytes and has been localized to presumptive mineralization zones of hyaline cartilage; collagens.

## Data Availability

Links to the OMNIBUS repository for RNA expression array studies analyzed: https://www.ncbi.nlm.nih.gov/geo/query/acc.cgi?acc=GSE103236; https://www.ncbi.nlm.nih.gov/geo/query/acc.cgi?acc=GSE13911; https://www.ncbi.nlm.nih.gov/geo/query/acc.cgi?acc=GSE79973.

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
