# Peer review of "Collagen Family and Other Matrix Remodeling Proteins Identified by Bioinformatics Analysis as Hub Genes Involved in Gastric Cancer Progression and Prognosis"

_ijms, 2022, doi:10.3390/ijms23063214_

Round 1

Reviewer 1 Report

The study highlighted, mainly associated genes coding proteins from the collagen family that can be used as important biomarkers that predict poor prognosis in gastric cancer.

The analysis is well performed and offers considerable insights however it is noted that COL12A1 (collagen type XII alpha 1 chain) and COL5A2 (collagen alpha-2(V) gene), Collagen Type X Alpha 1 Chain (COL10A1) in particular are related to Ehlers-Danlos syndrome. 

Ehlers-Danlos Syndrome Type IV is the only one currently to be related to Gastric Adenocarcinoma and is correlated with mutations in the gene COL3A1 encoding for type III pro-collagen synthesis.
From the results obtained, therefore, it is assumed that all patients with Ehlers-Danlos Syndrome should develop gastric cancer, and currently, there is no data.

It is recommended to:
- reassess of bioinformatics data according to this evidence.
- check the expression of COL3A1
- modify the title and discussion.

Author Response

We thank the reviewer for the comments. Indeed, UALCAN analysis shows that COL3A1 gene expression is increased in gastric tumor tissue compared to normal tissue and is correlated with survival. Moreover, other genes encoding proteins from the collagen family have increased expression in cancers, including gastric cancer.

We carefully re-examined the bio-informatics analysis looking for COL3A1, but it was not found on the list of the 22 common up-regulated genes obtained by running the Venn diagram tool that formed the basis of subsequent studies (please see the attachment). Moreover, network string interaction analyzed in Cytoscape was used to select only the genes with an MCODE score > 4 (as can be seen in the print screen below). Consequently, other genes from the collagen family (COL1A1, COL1A2) were left out of this analysis, the aim of our study being to retain a gene signature as predictive and significant as possible, having the highest interaction as a group.

 The pathology of Ehlers-Danlos syndrome involves misfolding and loss of function mutations in collagen genes (COL3A1, COL1A1, COL5A2) that result in a defective protein, and also structural mutations that reduce protein secretion (PMID: 30768852). Indeed, according to recent studies, misfolded somatic mutations in collagen genes may occur in gastric cancer similar to germline mutations from collagenopathy (eg Ehlers-Danlos syndrome), which leads to disrupted extracellular matrix, and are in strong association with overall survival. However, according to a recent study conducted by Brodsky A et al. (PMID: 35120467) collagen mutations were associated with collagen fibril decrease and better outcome while collagen overexpression was associated with poor outcome: “Truncation and missense collagen mutants reorganize the tumor microenvironment decreasing multiple processes that increase drug sensitivity and reduce metastasis risk including reduced EMT, less local collagen around the cancer cells, a more disorganized collagen structure, and increased infiltration of cytotoxic immune cells and drugs”.

We have included a comment to this effect in the discussions (page 11): “A similar degradation of the extracellular matrix is found in some genetic diseases (eg Ehlers-Danlos syndrome) associated with mutations in collagen genes. Thus, a recent study looked at the incidence of mutations in collagen genes and their role in gastric tumor progression, as well as their association with survival. The results showed that the mutations were associated with a distinctive lower matrisome expression, due to loss of collagen expression and secretion, strongly associated with improved outcomes (PMID: 35120467)”.

We would like to point out that our study did not assessed collagen mutations but was focused mainly on the overexpression of collagens family members identified as hub genes and their association with overall survival, intending to identify genes that could be candidate biomarkers for gastric cancer progression and prognosis and new therapeutic targets.

Reviewer 2 Report

Some discussion on the reason why TIMP1 is correlated with poor survivals in GC patients may be added in introduction, and the relationships between previous findings and the results in terms of tissue reorganization may be added in discussion. 

Author Response

We thank the reviewer for comments and suggestions.

Regarding TIMP1, we added a phrase in the introduction section (page 2) explaining why TIMP1, an MMP inhibitor, could act as a tumor promotor being associated with poor prognosis: “Although TIMP1 is a tissue inhibitor of metalloproteinases, it may also have MMP-independent functions in solid cancers. TIMP-1 can stimulate cell proliferation, accelerating tumor invasion and metastasis, via important signaling pathways such as NOTCH and WNT (PMID: 27932800, PMID: 33628641), recent reports demonstrating the poor prognostic value of TIMP-1-positive expression in solid cancers [PMID: 31578137].  

Moreover, one paragraph was added in the discussion section regarding relationships between previous findings and our results in terms of tissue reorganization during tumor progression (page 11-12-highlighted in red). This comes in addition to the existing paragraph regarding collagens, tissue reorganization, and desmoplasia on the same page.

Reviewer 3 Report

In this manuscript, the authors re-analyzed published microarray data using the publicly-available bioinformatic tools to predict whether expression of extracellular matrix genes may correlate with cancer progression. The authors validate their findings by performing Western blotting using their own clinical samples. It is not surprising that the expression of extracellular matrix (both at the levels of mRNAs and proteins) change in tumors compared to healthy tissues as the tissue/cellular composition is different. Thus, it is nothing new to be presented here. More specific comments are listed below:

Major points:

[1] Lines 70 - 73: "To this end, we used our previous data GSE1032364 together with other two gene expression microarray datasets from the Gene Expression Omnibus (GEO) (GSE139112, GSE799733), and several bioinformatics tools, to identify aberrant expressed genes significantly involved in gastric carcinogenesis and progression."

When the above mentioned GSE IDs were searched, only GSE139112 exist. Also, GSE139112 contain ChIP-seq data; NOT microarray data.

GSE1032364: Not existing

GSE139112: ChIP-seq analysis of dCas9 chromatin occupancy in K562 cells by CAPTURE2.0

GSE799733: Not existing

What did the authors analyze in this study?

[2] Lines 85 - 86: "Gene expression was analyzed by the GEO2R tool, with log FC ≥2, and p<0.05 as standards to identify DEGs." The p-values must be corrected with multiple test, such as false discovery rate (FDR)?

[3] Lines 202 - 206: "Next, protein expression was tested in clinical specimens of tumor and normal gastric tissue through western-blot assay (Figure 4B). Results showed an increase of COL10A1 protein in tumor tissue, and in BGN and FAP proteins involved in collagen fibril assembly and matrix degradation, respectively, emphasizing that these hub genes could have a crucial function in gastric tumorigenesis." Although the authors validate mRNA expression with Western blotting assay for protein expression, the authors do not discuss about how tissues differ from tumor and normal gastric samples. The authors must provide immuno-staning of samples as it is most likely that the changes that they observed are due to cell-type differences.

The above point is especially important as the authors conclude as follows in the next subsection:

Lines 233 - 236 "However, a positive and significant correlation was noticed with the presence of macrophages, the predominant type being the immunosuppressive M2 macrophages (R value 0.255 – 0.448, p-235 value range 4.82e-07-1.63e-13) which predicts a poor prognosis (Table S1).”

Minor points:

(1) Why some sentences are in bold? For example, "Gastric cancer remains the top five cancers both in terms of incidence and mortality."

Author Response

Point 1: Lines 70 - 73: "To this end, we used our previous data GSE1032364 together with other two gene expression microarray datasets from the Gene Expression Omnibus (GEO) (GSE139112, GSE799733), and several bioinformatics tools, to identify aberrantly expressed genes significantly involved in gastric carcinogenesis and progression."

 When the above mentioned GSE IDs were searched, only GSE139112 exist. Also, GSE139112 contain ChIP-seq data; NOT microarray data.

 GSE1032364: Not existing

GSE139112: ChIP-seq analysis of dCas9 chromatin occupancy in K562 cells by CAPTURE2.0

GSE799733: Not existing

What did the authors analyze in this study?

Response 1: We thank the reviewer for the observation and apologize for this typing error that appears only in the introduction section. We’ve corrected the ID numbers for the GSE datasets according to those from Materials and Methods section 2.1 (page 2). Microarray dataset information and figure 1. Please find below links to the OMNIBUS repository for each of them. All array studies are based on total RNA expression profiling.

https://www.ncbi.nlm.nih.gov/geo/query/acc.cgi?acc=GSE103236

https://www.ncbi.nlm.nih.gov/geo/query/acc.cgi?acc=GSE13911

https://www.ncbi.nlm.nih.gov/geo/query/acc.cgi?acc=GSE79973

Point 2: Lines 85 - 86: "Gene expression was analyzed by the GEO2R tool, with log FC ≥2, and p<0.05 as standards to identify DEGs." The p-values must be corrected with multiple test, such as false discovery rate (FDR)?

Response 2: We thank the reviewer for the helpful suggestion. We have added in the Materials and Methods section 2.1. Microarray dataset information the information about the test used for correction of the p-value (page 2): “The Benjamini and Hochberg false discovery rate method was applied for Geo2R analyses”.

 Point 3: Lines 202 - 206: "Next, protein expression was tested in clinical specimens of tumor and normal gastric tissue through western-blot assay (Figure 4B). Results showed an increase of COL10A1 protein in tumor tissue, and in BGN and FAP proteins involved in collagen fibril assembly and matrix degradation, respectively, emphasizing that these hub genes could have a crucial function in gastric tumorigenesis." Although the authors validate mRNA expression with Western blotting assay for protein expression, the authors do not discuss about how tissues differ from tumor and normal gastric samples. The authors must provide immuno-staning of samples as it is most likely that the changes that they observed are due to cell-type differences.

 The above point is especially important as the authors conclude as follows in the next subsection:

 Lines 233 - 236 "However, a positive and significant correlation was noticed with the presence of macrophages, the predominant type being the immunosuppressive M2 macrophages (R value 0.255 – 0.448, p-235 value range 4.82e-07-1.63e-13) which predicts a poor prognosis (Table S1).”

Response 3: We thank the reviewer for the comments and helpful suggestions. We have included a comment in the results section discussing the differences between tumor and normal gastric tissue (page 8): “The results show an abundance of ECM proteins, like collagens, and remodeling enzymes (BGN, FAP) in tumor tissue, compared to the normal one. These proteins are secreted mainly by cancer-associated fibroblast and infiltrating immune cells”.

We considered tumor tissue as a whole, a heterogeneous collection of infiltrating and resident host cells (tumor, stromal, blood vessel), and our study focused on identifying biomarkers associated with tumor progression, rather than identifying localization and distribution of these proteins at the tissue level.

Point 4: Why some sentences are in bold? For example, "Gastric cancer remains the top five cancers both in terms of incidence and mortality."

Response 4: Thank you for your observation. We wanted to highlight some ideas. We restored it to the normal format.

Round 2

Reviewer 1 Report

The adjustments introduced to the text and the clarifications made have improved the clarity of the article, bolstering the observations. There are no more adjustments to be made.

Reviewer 3 Report

I have no further comment to make.